# Prevalence and detection of low-allele-fraction variants in clinical cancer samples

Hyun-Tae Shin[1,2], Yoon-La Choi[2,3], Jae Won Yun[1,2], Nayoung K.D. Kim[1], Sook-Young Kim[1], Hyo Jeong Jeon[1], Jae-Yong Nam[1,2], Chung Lee[1,2], Daeun Ryu[1,2], Sang Cheol Kim[1], Kyunghee Park[1], Eunjin Lee[1], Joon Seol Bae[1], Dae Soon Son[1], Je-Gun Joung[1], Jeeyun Lee[4], Seung Tae Kim[4], Myung-Ju Ahn[4], Se-Hoon Lee[4], Jin Seok Ahn[4], Woo Yong Lee[5], Bo Young Oh[5,6], Yeon Hee Park[4], Jeong Eon Lee[5], Kwang Hyuk Lee[7], Hee Cheol Kim[5], Kyoung-Mee Kim[3], Young-Hyuck Im[4], Keunchil Park[4], Peter J. Park [8] & Woong-Yang Park [1,2,9]

Accurate detection of genomic alterations using high-throughput sequencing is an essential component of precision cancer medicine. We characterize the variant allele fractions (VAFs) of somatic single nucleotide variants and indels across 5095 clinical samples profiled using a custom panel, CancerSCAN. Our results demonstrate that a significant fraction of clinically actionable variants have low VAFs, often due to low tumor purity and treatment-induced mutations. The percentages of mutations under 5% VAF across hotspots in *EGFR*, *KRAS*, *PIK3CA*, and *BRAF* are 16%, 11%, 12%, and 10%, respectively, with 24% for *EGFR* T790M and 17% for *PIK3CA* E545. For clinical relevance, we describe two patients for whom targeted therapy achieved remission despite low VAF mutations. We also characterize the read depths necessary to achieve sensitivity and specificity comparable to current laboratory assays. These results show that capturing low VAF mutations at hotspots by sufficient sequencing coverage and carefully tuned algorithms is imperative for a clinical assay.

[1] Samsung Genome Institute, Samsung Medical Center, Seoul, 06351, Korea. [2] Samsung Advanced Institute of Health Science and Technology, Sungkyunkwan University, Seoul, 06351, Korea. [3] Department of Pathology and Translational Genomics, Samsung Medical Center, Sungkyunkwan University School of Medicine, Seoul, 06351, Korea. [4] Department of Hematology and Oncology, Department of Medicine, Samsung Medical Center, Sungkyunkwan University School of Medicine, Seoul, 06351, Korea. [5] Department of Surgery, Samsung Medical Center, Sungkyunkwan University School of Medicine, Seoul, 06351, Korea. [6] Department of Surgery, Ewha Womans University School of Medicine, Seoul, 07985, Korea. [7] Division of Gastroenterology, Department of Medicine, Samsung Medical Center, Sungkyunkwan University School of Medicine, Seoul, 06351, Korea. [8] Department of Biomedical Informatics, Harvard Medical School, Boston, MA 02115, USA. [9] Department of Molecular Cell Biology, Sungkyunkwan University School of Medicine, Seoul, 16419, Korea. Hyun-Tae Shin, Yoon-La Choi, Jae Won Yun and Nayoung K.D. Kim contributed equally to this work. Correspondence and requests for materials should be addressed to P.J.P. (email: peter_park@hms.harvard.edu) or to W.-Y.P. (email: woongyang.park@samsung.com)

Genomic profiling of tumors by high-throughput sequencing has fueled rapid progress on our understanding of the molecular features underlying all steps of carcinogenesis—tumor initiation, progression, response to treatment, and relapse[1,2]. Sequencing technology continues to advance quickly, notably with whole-genome sequencing (WGS) becoming more affordable, and amplification and sequencing of RNA and DNA at the single cell level becoming possible[3,4]. However, translation of the insights from molecular profiling to patient care has been much slower. Several studies have shown that selection of therapy based on genomic profiling of few hotspot mutations could lead to prolonged survival for patients[5–7]. But the repertoire of drugs available for treatment has not been expanding as rapidly as our ability to identify the mutation, limiting the usefulness of whole-exome sequencing (WES) or WGS.

A number of factors are prerequisite for a successful implementation of a genome-guided therapy selection in routine cancer care. First, obtaining a representative tumor specimen of sufficient quality for genome profiling is an on-going challenge. Given the heterogeneity of a tumor in an individual[8], a full description of the tumor may require multiple samplings of different geographic regions, but this is not feasible in the clinic. Profiling of circulating tumor cells offers a promising approach for a non-invasive and serial characterization, but it is limited to a minority of tumor types and relies on a less mature technology and extremely deep sequencing[9]. Second, a comprehensive profiling of mutations in a tumor is possible but requires multiple assays and extensive bioinformatic analysis. The assay platform ranges from full genome coverage (typically at 30–60×) with WGS to exome-only coverage (typically 100-200×) to narrow genome coverage (typically at 500–1000×) with panel sequencing. These platforms offer a trade-off between the depth and breadth of profiling. Sequencing RNA is also an attractive option, as it can be used to identify single nucleotide variants (SNVs)/insertions and deletions (indels) and gene fusions from expressed transcripts. All these assays are complimentary and can be useful for cross-validating mutations, but performing multiple assays is often prohibitive due to the cost and the lack of expertise needed for data analysis. Third, even when a somatic mutation is found, assessing whether it plays a role as a 'driver' rather than a 'passenger' is difficult. A variety of resources such a catalog of previously observed somatic mutations[10] and computational tools for predicting the effect of an observed coding mutation on protein function exist[11,12]. However, a large integrated database of cancer genome profiling and clinical data to infer whether a mutation might be correlated with treatment response is lacking. At this time, such information is available only for a relatively small set of hotspot mutations.

As the catalog of hotspot mutations that are associated with clinical outcome and the drugs that target those mutations continue to expand, a critical area of improvement is to increase the sensitivity of detecting known hotspot mutations. Here we focus on the questions of how reliably we can detect somatic mutations in cancer using a targeted sequencing panel—especially in comparison to the currently available clinical assays—and whether hard-to-detect mutations that require high sequencing coverage are clinically relevant. The first question is intimately related to the question of how the variant allele fractions (VAFs) for such mutations are distributed in a large population of cancer patients, since it is difficult to assess what the appropriate sequencing

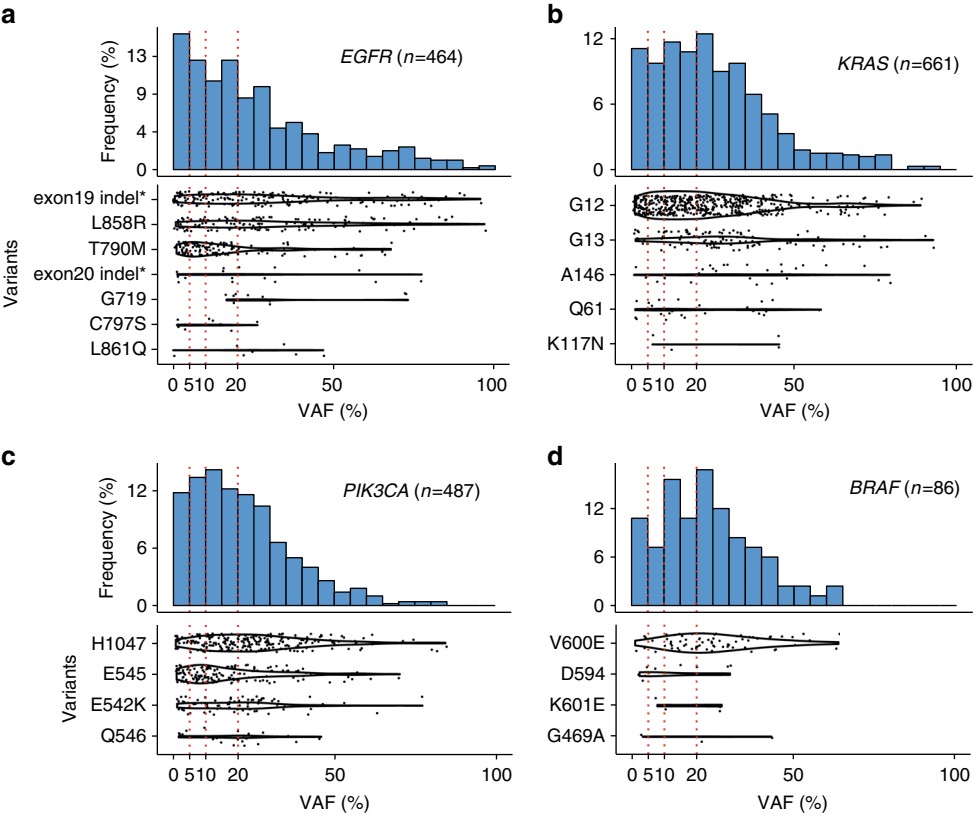

**Fig. 1** Prevalence of point mutations with low variant allele fraction (VAF) in cancer specimens. VAF distributions of the four most frequently mutated actionable genes in our data: *EGFR*, *KRAS*, *PIK3CA*, and *BRAF*. Each dot corresponds to a sample, with the violin plot showing the estimated density; the red vertical dotted lines are at 5, 10, and 20% VAF. 24% of the *EGFR* T790M, 17% of *PIK3CA* E545, and 12% of *KRAS* G12 mutations are below 5%. Two important non-frameshift indels (indicated by asterisk) are also included for *EGFR*

coverage is unless we have the VAF information; the second question requires curation of treatment response data for a large number of cases. To address these issues, we have performed detailed analysis of 5095 clinical cancer samples profiled at the Samsung Medical Center over the past three years. Our analysis details the observed mutations in a large cohort, describes the VAF distribution at hotspot mutations, estimates the sequencing depth necessary to obtain the "limit-of-detection" that is similar to what is standard for clinical assays, reports the estimated distribution of tumor purity across the cohort, describes the issues related to formaldehyde fixed-paraffin embedded (FFPE) tissues, gives examples of cases in which low VAF mutations were used to select treatment for patients, and compares the progression free survival data for patients with a low VAF vs high VAF variant who received targeted therapy.

## Results

**Sequencing of cancer patients.** Over a period of three years, we sequenced DNA from 5095 patients who could potentially benefit from a clinical trial if an actionable mutation is discovered. We utilized a custom-designed panel that covered up to 381 cancer-related genes curated from the literature (Supplementary Data 1), as well as the *TERT* promoter and introns that contain frequent breakpoints for selected fusion candidates. These selected genes covered variants associated with the targeted cancer therapies approved by the US Food and Drug Administration (FDA), Korean Ministry of Food and Drug Safety (MFDS) or in the clinical trials at the Precision Oncology Clinic at Samsung Medical Center. A custom panel offered the flexibility to rapidly incorporate the relevant candidate variants as new clinical trials are initiated. About half of the cases were also refractory tumors with a history of recurrence and/or metastasis (Supplementary Data 2); nearly half of the samples were formalin-fixed paraffin-embedded (FFPE) tissue samples. Most samples had the mean coverage of ~900× (Supplementary Data 3), with the coverage at hotspots well above the mean.

**Characterization of mutant allele frequencies at hotspots.** After standard quality control and genome alignment, we identified SNVs using a combination of two algorithms, MuTect[13] and LoFreq[14]. The former implements a Bayesian classifier with several post-calling filters to reduce false positives; the latter was designed specifically for extremely low VAF variants. We further designed a filtering step based on a regression model trained on low VAF variants in normal samples and applied it to somatic variant calls. The union of the two callers and the additional filtering step substantially increased the accuracy of the variant list, according to our tests on simulation data consisting of mixtures of well-characterized samples at specified ratios (More details are in Methods section).

We did not have a paired normal tissue in the majority of the cases, so we used a set of >400 normal samples with matched ethnicity to remove germline variants (Methods section). To test the effectiveness of this filtering process, we sequenced 74 breast cancer samples along with their paired normals (tumor: CancerSCAN V1, 83-gene panel at ~900×, normal: WES at ~100×) and examined how the filtering using a panel of normals compares with using paired normals. We found that the vast majority (94.5%; 1928 out of 2040) of the germline variants identified from paired WES data were removed by the panel of (unpaired) normals. This is consistent with the recent paper showing that, with ~400 unrelated genomes for filtering germline variants, somatic SNVs and indels could be distinguished from germline ones as efficiently as having the matched normals[15].

Importantly, we found that the VAFs at many clinically-important hotspots are often very low across our samples. In Fig. 1a–d, we show the distribution of VAFs for the four genes with the highest frequency of SNVs in our data; other genes are shown in Supplementary Fig. 1. Across the 20 hotspots, *EGFR*, *KRAS*, *PIK3CA*, and *BRAF* have 28, 21, 26, and 17% of mutations under 10%, respectively. The percentages of mutations under 5% VAF are 16, 11, 12, and 10%, respectively. To give examples of specific hotspots, 24% of the *EGFR* T790M, 17% of *PIK3CA* E545, and 12% of *KRAS* G12 mutations are below 5%. These hotspot SNVs were covered at an average depth of 1151×. To confirm that the VAFs estimated from the sequencing data are accurate, we compared the VAFs obtained from sequencing with those estimated from digital PCR (dPCR) and found that they have high correlation, as shown in Fig. 2a and Supplementary Data 4.

Variants in *EGFR* exemplify the importance of detecting low VAF cases. In 40-50% of Asian patients and 10% of white patients with non-small cell lung cancer (NSCLC), somatic mutations are present in *EGFR*[16]. Treatment with EGFR- tyrosine kinase inhibitors (TKI) such as erlotinib, gefitinib, and afatinib in those patients have shown improved outcomes compared to conventional chemotherapy, with a response rate of 50–75%[17]. However, >50% of the cases, the patient will develop acquired resistance within 1 or 2 years in the form of a second-site mutation, *EGFR* T790M, in the *EGFR* kinase domain. For these patients, osimertinib (AZD9291), an oral irreversible EGFR-TKI (third generation EGFR-TKI), has been shown to be effective with improved safety profile[18], and was approved in 2015 by the US FDA. This makes the detection of this variant critical for patients with T790M-mediated resistance to EGFR-TKI. In our cohort, this variant was present in 113 cases, and 24% had VAF of under 5%.

Furthermore, a study identified another acquired mutation *EGFR* C797S that confers resistance to AZD9291[19]. Successful molecules for overcoming this mutation were reported, too[20]. In our cohort, we identified eight lung cancer patients with *EGFR* C797S mutation occurring in *cis* with *EGFR* T790M mutation (Fig. 2b–e and Supplementary Fig. 2); in one patient, serial sampling confirmed that the *EGFR* C797S mutation occurred after the AZD9291 therapy (Fig. 2e). Since *EGFR* C797S appears as an acquired resistance to target therapy, its VAFs were even lower than that of *EGFR* T790M mutation or *EGFR* activating mutations in the 8 samples, with the four cases having VAF of 1.4, 2.2, 3.9, and 4.0% (Fig. 2b). These findings highlight the need for a platform with high sensitivity for low VAF variants.

These examples hint at the possibility that the VAFs in post-chemotherapy samples may be lower in general. To determine whether this is the case, we were able to classify 1557 samples as pre- or post-treatment (1203 vs 354, respectively) based on clinical chart review (Supplementary Data 2). We find that the overall VAF distributions of hotspot mutations tend to be a little lower in the post-treatment samples but do not reach statistical significance (Supplementary Fig. 3). Similarly, tumor purity also did not show statistically significant difference between the pre- and post-treatment samples. In some specific cases, e.g., in the post EGFR-TKI therapy samples, we expect to find tumor resistance-inducing clones containing secondary mutations after treatment. In the 141 lung cancer cases divided into pre- ($n = 31$) and post- ($n = 110$) EGFR-TKI therapy, we found that the VAF distributions at the hotspots were significantly different ($P = 0.018$), mostly due to *EGFR* T790M (Fig. 2f). The mutation counts, at least as represented on the capture target area (CancerSCAN V2 panel only), did not display consistent change between pre- and post-treatment in the several cancer types (Supplementary Fig. 4).

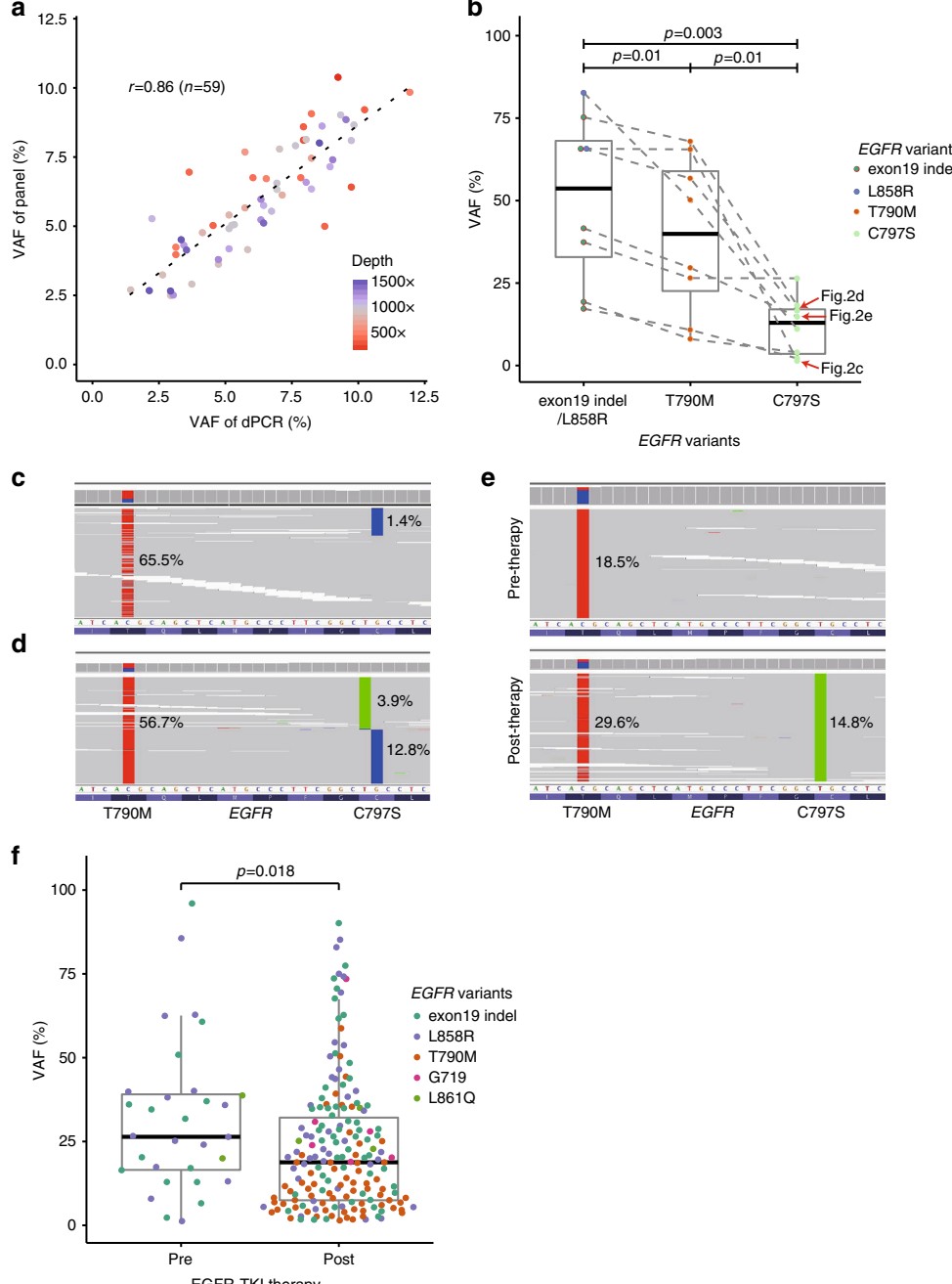

**Fig. 2** Features of actionable mutations with low VAF. **a** Concordance between VAFs estimated from panels and dPCRs for low VAF variants. The Pearson correlation coefficient based on 59 actionable variants is 0.86. Variants with higher coverage (colors correspond to sequencing depths) tend to show higher correlation; also see Supplementary Data 4. **b** Differences in VAF distributions among *EGFR* mutations in eight refractory lung cancer samples harboring *EGFR* C797S. All samples had an activating mutation (*EGFR* exon19 non-frameshift (NFS) deletion or L858R) and two resistance mutations (*EGFR* T790M and C797S), with the latter occurring at lower VAF. Dotted lines indicate mutations belonging to the same sample. The *P*-values were calculated using the paired *t*-test. **c**, **d** Browser view of the case in which *EGFR* C797S occurred in *cis* with *EGFR* T790M (only a subset of the reads are shown). **e** An acquired *EGFR* C797S mutation is found after AZD9291 therapy. **f** Comparison of VAF of *EGFR* variants of lung cancer samples (n = 141) pre- and post-EGFR-TKI therapy (afatinib, erlotinib, or gefitinib). The *P* values were calculated using the Wilcoxon rank sum test

**Determining sufficient sequencing coverage.** For a test based on high-throughput sequencing to be adopted in clinical practice, the sensitivity and specificity of the test must be at least on par with the currently available single-gene based tests. In conventional molecular tests in the clinic, a standard metric for the performance of a test is the limit-of-detection (LOD), the lowest concentration of an analyte that can be detected reliably (typically defined as having 95% sensitivity)[21]. A guideline by the American

College of Medical Genetics and Genomics for next-generation sequencing-based assays does not recommend a specific coverage threshold because the sensitivity and specificity depend on several aspects of the assay and platform (e.g., base-call error rates, allelic bias, presence of duplicate reads, and the performance of the analytical pipeline). However, it recommends that a laboratory should note the percentage of bases that reach the desired minimum coverage in the target region, and recognizes the

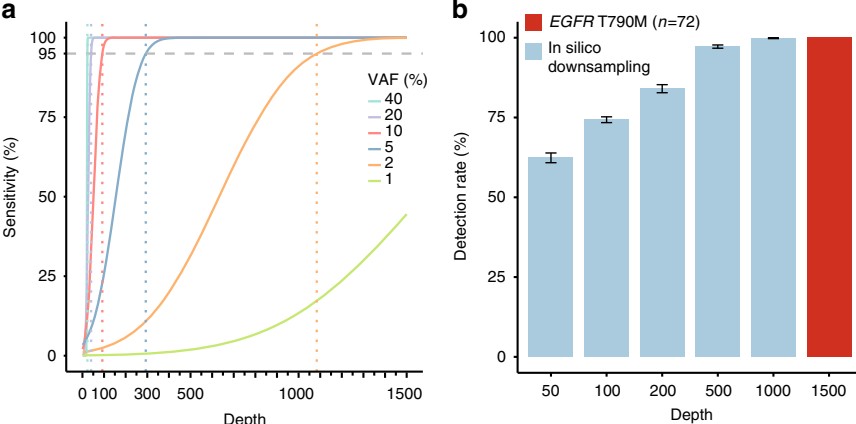

**Fig. 3** Impact of sequencing depth on single nucleotide variant (SNV) detection. **a** Limit of detection (LOD) is defined as the VAF for which 95% sensitivity is achieved for a given depth (Methods section). On the asis of subsampling analysis of computational mixture data, the LOD of 2% is achieved at 1085×, 5% at 294×, 10% at 94×, 20% at 40×, and 40% at 18×. **b** 72 samples harboring *EGFR* T790M were down-sampled from the original depth, and detection rates were measured at each depth (10 iterations), assuming that all variants were identified with the full data (1500×). Detection rate for this variant is <75% even for 100× data. Error bar: s.e.m

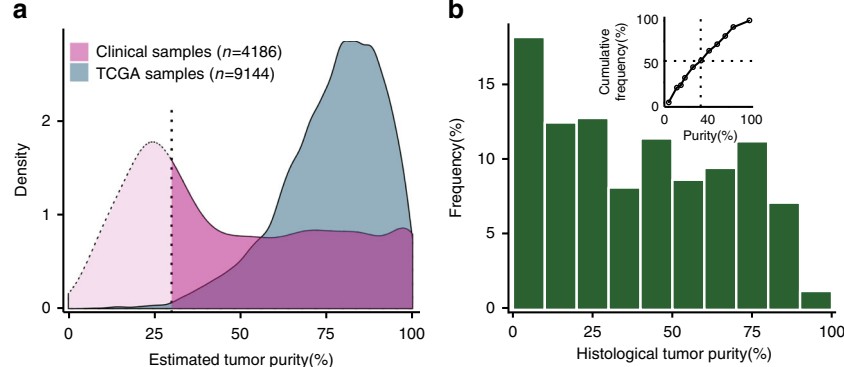

**Fig. 4** Distribution of tumor purity in clinical samples. **a** The distributions of tumor purity estimates in our cohort and the TCGA samples are shown after kernel density smoothing. It was possible to reliably estimate purity (Supplementary Data 3) for about half the cases in our cohort (Methods section); purity estimates for the TCGA samples are taken from Aran et al.[23]. Below 30%, the estimates are less reliable, as indicated by the dotted line. **b** Histological tumor purity estimates of 3697 lung cancer biopsy specimens obtained in daily clinical practice described in a related cohort[24]. The cumulative distribution (inset) shows that half of the samples had purity <40%

importance of each laboratory determining the lower LOD of variants based on dilution assays[22].

To determine LOD for our panel-based test, we examined the sensitivity of detection for a somatic SNV of given VAF as a function of sequencing depth, using both simulated and experimental data (Methods section). The simulation results using the well-annotated variants in the HapMap sample NA12878 show that depth of coverage needed to maintain a given sensitivity increases greatly as VAF decreases (Fig. 3a). For a 95% sensitivity and >95% positive predictive value (PPV; the fraction of true variants among all called variants), ~ 40× is needed for variants with 20% VAF, ~ 94× is needed for variants with 10% VAF, but ~ 294× is needed for 5% VAF and ~ 1085× is needed for 2% VAF. One reason for the high coverage needed (e.g., ~ 1000× for a 2% variant means ~ 20 reads containing the variant on average) is the high specificity required for this assay as well as our accounting of the sampling error (i.e., a true 2% VAF variant is present at variable percentages centered at 2%).

We also performed a re-sampling experiment with patient data as another way of examining the impact of sequencing depth on the detection rate. For the *EGFR* T790M mutation, present in 72 patients without *EGFR* amplification (Supplementary Data 5), subsampling from the real data shows that the average detection

rate of *EGFR* T790M (assuming that all mutations have been detected in our ~ 1500× data at this position) is 84% at 200×, 74% at 100×, and 62% at 50× (Fig. 3b). These results suggest that whole-exome sequencing, which typically has 100-200× coverage, may miss 15–30% of this actionable mutation.

In a typical variant report based on panel-based assays, the average coverage is given but not LODs for each gene or each position. Without an LOD, when a variant is *not* called at a given position, it is difficult to distinguish between the true absence of the variant and the lack of statistical power to detect one. Despite improvements in the target capture step, the coverage across a target region remains uneven for most platforms (Supplementary Fig. 5). Indeed, a recent paper showed the counter-intuitive result that WGS could be more powerful than exomes for detecting *exonic* variants, with 3% of variants detected on WGS missed on exomes due to uneven coverage. We thus advocate annotating LOD at the gene or the nucleotide level, at least for clinically-relevant mutations. One possibility is to do it graphically, denoting LOD using the background color (Supplementary Fig. 6 and Supplementary Data 6), but there may be other ways to incorporate this information. Importantly, we espouse the use of LOD as the more useful metric than sequencing depth alone, since LOD incorporates additional

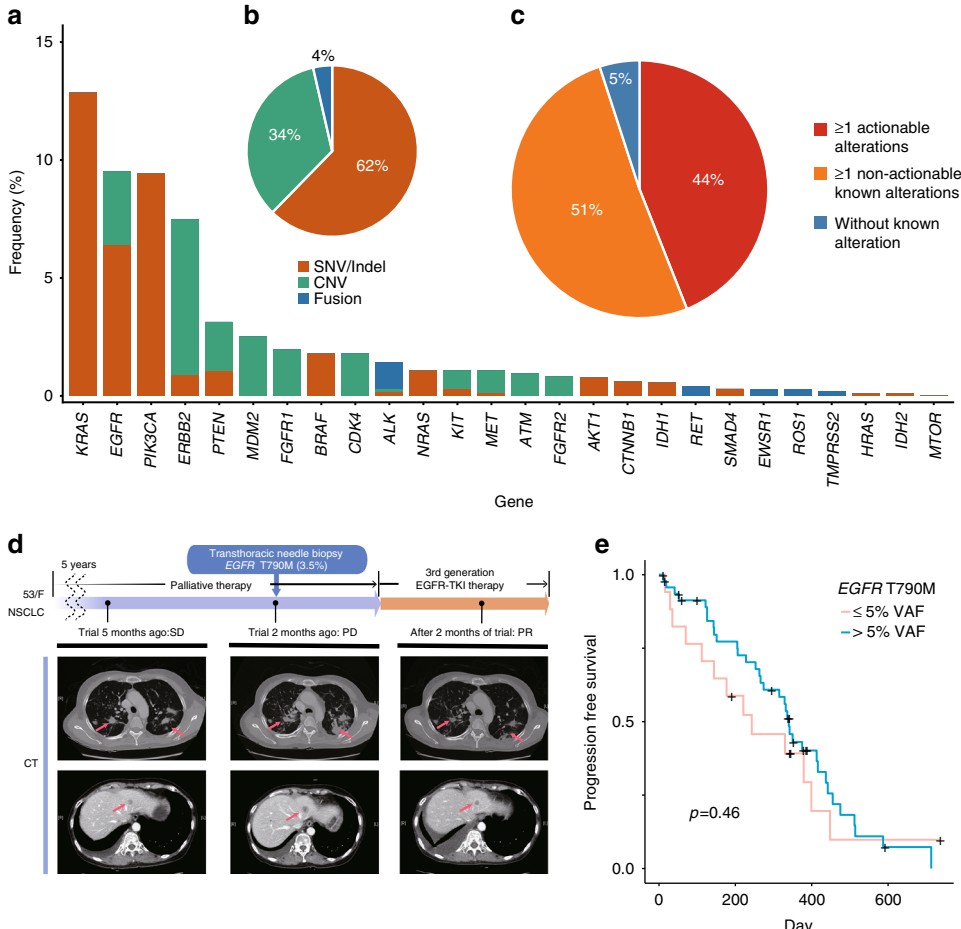

**Fig. 5** Variant classification and illustration of clinically-relevant low VAF mutation. **a** Frequency of genes with actionable alterations, colored by alteration type. **b** Distribution of SNVs/indels, CNVs, and fusions. **c** Proportions of patients with actionable alterations, non-actionable but known alterations (as catalogued by COSMIC), and without known alterations, among the samples sequenced by CancerSCAN V2 ($n = 3598$). **d** Examples of a metastatic lung cancer patient harboring low VAF (3.5%) *EGFR* T790M who had partial remission (PR) to the targeted therapy. **e** Comparison of the progression-free survival curves for lung cancer patients ($n = 65$) receiving third generation EGFR-TKI therapy with *EGFR* T790M. There is no statistically significant difference in survival between those with high and low VAF variants. SD, stable disease; PD, progressive disease

information on the quality of the data and the characteristics of the analytical pipeline.

**Impact of tumor purity**. The issue of what constitutes sufficient sequencing depth is influenced by other characteristics of the patient sample, including tumor purity and the clonality of the mutations, as well as the tissue preparation method and the platform used. Lower tumor purity proportionally reduces the effective coverage of the variant alleles in tumor cells, reducing detection sensitivity. Importantly, whereas research projects may impose a minimum threshold for tumor purity at the sample selection stage, there is little control over tumor purity in the clinic. In particular, many clinical specimens are derived from biopsies rather than surgical resections, and the tumor purity in such cases may be very low.

Although purity is difficult to estimate from panel data, we were able to obtain estimates in about half of the cases (Methods section). The distributions of purity estimates for our samples and the TCGA samples[23] showed a striking difference (Fig. 4a). As an additional piece of evidence, in ~3600 lung biopsy specimens from a separate study we conducted, the histological purity estimates are ≤20% in 30% and ≤40% in 51% of the cases (Fig. 4b)[24].

Furthermore, intra-tumor heterogeneity can result in additional variability that necessitates higher coverage. For example, even if the mutation is present in 10% in the entire tumor, the VAF of the biopsied portion may only be 2% due to spatial heterogeneity. Another challenging aspect of working with clinical samples is that they are often formalin-fixed paraffin-embedded (FFPE) tissues with degraded DNA and smaller fragment sizes[25]. As a result, the effective coverage of an FFPE sample after removing duplicate reads is substantially lower than the target coverage (~800× vs ~1,000× in our data). There is, however, a significant gene-to-gene variation, with the coverage at some genes higher in FFPE samples than in fresh frozen samples (Supplementary Fig. 7).

**Frequency of alterations at actionable mutations**. Across the 5095 cases, the genes most frequently altered with actionable mutations are shown in Fig. 5a, starting with *KRAS* (13%), *EGFR* (9.5%), *PIK3CA* (9.4%), and *ERBB2* (7.5%). In addition to SNVs, we detected copy number variations (CNVs) and common fusions using custom algorithms (Methods section). Overall, SNVs/indels accounted for 62% of the detected variants while CNVs and fusions accounted for 34% and 4%, respectively, spread across multiple tumor types (Fig. 5b). About 44% of the

patients had "actionable" alterations, i.e., the choice of treatment regimen was based at least in part on the presence of the mutation (Fig. 5c and Supplementary Fig. 8). In another 51%, the observed mutation was present in the Catalogue of Somatic Mutations in Cancer (COSMIC) but no treatment was available at present. These numbers suggest the panel-based testing for cancer patients is informative for nearly half the patients tested and that the number of will continue to increase as more drugs become available. These estimates are similar to the numbers reported in other studies[26].

**Clinical relevance of low VAF variants**. In primary tumors, a 'driver' mutation is expected to have a relatively high VAF due to their occurrence early in carcinogenesis and the selective advantage conferred by the mutation; in refractory tumors, however, a secondary driver mutation that arises in response to a treatment may be present with a low VAF[27,28], as we showed above. To illustrate the importance of such mutations in patient care, we describe two examples of refractory cancer patients who had clinical response to targeted therapy selected based on a low VAF mutation.

The first case is a 59-year-old never-smoker female with NSCLC with multiple distant metastasis including bone and liver (Fig. 5d). She underwent a lobectomy, but the cancer recurred; in spite of palliative therapy (conventional chemotherapy, *EGFR* targeted therapy, and radiotherapy), her cancer progressed slowly for 5 years. When her condition worsened, she underwent a transthoracic needle aspiration biopsy. Genomic profiling of the biopsied specimen revealed *EGFR* L858R (29%) and T790M (3.5%). The patient was subsequently enrolled with an AZD9291 trial and achieved partial remission. The second case is a 70-year-old female patient with metastatic gastric cancer with peritoneal seeding (Supplementary Fig. 9). After failing 8 cycles of capecitabine/oxaliplatin chemotherapy, she underwent an esophagogastroduodenoscopy biopsy. Genomic profiling of the biopsied tissue revealed a *PIK3CA* E542K with 4.1% VAF, which was confirmed by dPCR (Supplementary Data 4). As a result, the patient was treated with an AKT inhibitor and has achieved partial remission, although the response was not durable.

Another approach to probe clinical relevance of low VAF variants is to compare survival data of patients with low *vs* high VAF variants who receive targeted therapy. If the low VAF variants are as important as the high VAF ones, we expect to see little difference in the survival curves; on the other hand, if the low VAF variants are not informative, we expect to see significant differences. Examining lung cancer patients ($n = 65$) with *EGFR* T790M who received 3rd generation EGFR-TKI, we find that the Kaplan-Meier curves for progression-free survival do not show significant differences between the low VAF cases ($\leq 5\%$) and the high VAF cases (>5%) (Fig. 5e). To show that this result does not depend on the specific threshold for defining "low" vs "high" allele fraction, we also tried all possible subdivision into two groups and computed the log-rank test in each case. We observed that none of the *P* values were significant (Supplementary Fig. 10).

## Discussion

Our analysis of 5,095 clinical samples demonstrates that a significant portion of clinically-relevant mutations are present at lower VAFs than some may have assumed. There are two primary factors for the lower VAFs. First, a substantial fraction of our cohort comprised refractory cases that had treatment-induced secondary mutations, which are more likely to have lower VAFs. Second, the estimated tumor purities in our samples were frequently low, causing the observed VAFs to be decreased by the

same percentage. It is noteworthy that the clinical specimens obtained at our hospital were mainly from small biopsy specimen, so had generally lower tumor purity than those observed in TCGA samples, which were collected from operation specimen for multi-dimensional profiling (DNA sequencing, RNA-seq, SNP arrays, DNA methylation, proteomics, etc.). Given that TCGA data have been used for developing and testing many algorithms for estimating purity and somatic variant calling, those algorithms may not be optimally tuned for some small-sized clinical samples or damaged FFPE DNA.

The presence of many low VAF mutations has implications for platform design. Targeted sequencing allows for the most sensitive characterization of SNVs/indels in a subset of genes, but cannot detect other variants such as gene fusions unless there are specifically targeted with additional probes (we showed that it is at times possible to identify translocations when discordant or split reads are pulled down in the capture step[29], but the sensitivity is too low in general). With our CancerSCAN V2 panel data, we have found that purity estimation was possible for about half of the samples in which the target regions included multiple CNVs and tumor purity was not too low. Our previous comparison of exome CNV identification algorithms revealed highly discordant result among six popular algorithms[30]; however, we were able to call CNVs reliably whenever the magnitude of the CNV was large and the purity was not too low. WGS allows for less sensitive characterization of SNVs but across the entire genome, including non-coding regions, and it can be used to accurately identify copy number variants and various rearrangements with nucleotide resolution. With an increasing amount of available functional genomics and epigenomics data, we are starting to recognize the role of some non-coding variants, e.g., mutations in enhancers that mediate the activity of tumor suppressor or oncogenes[31]. Nonetheless, our results make it clear that very high-depth sequencing on a custom panel should be the first assay in the clinic, with flexibility to modify the targeted list as new drugs become available through clinical trials for specific mutations. Indeed, it appears that even exome sequencing, typically at 100–200×, would not have sufficient sensitivity unless the coverage is increased several-fold.

One limitation of this study is that our analysis was focused on mutational hotspots. When low VAF mutations are observed at other sites, it becomes more difficult to determine whether they are true mutations or are artifacts due to misalignment, sequencing error, or extraneous biological sources (e.g., oxidative damage or formalin fixation in sample preparation). To remove such artifacts, one needs a sophisticated variant calling method that retains high specificity even for low VAF cases. Random sequencing errors occurring at the same nucleotide position in multiple reads is vanishingly small, and we should be able to identify some variants even with 3 or 4 reads (at 1000×, 0.3–0.4%) if there are appropriate filters to remove less confident cases[13]. Other technical artifacts, e.g., due to read mis-alignment, can be avoided by annotating alignment error-prone regions from a large set of panel data. FFPE-induced mutations and other biological artifacts are more challenging to distinguish, as there are currently no high-resolution models to predict where such mutations are likely to arise at a given location.

There are several other challenges in panel sequencing. A recent paper on the effect of tumor sequencing without the matched normal suggested that up to a third of actionable changes might be incorrectly classified as somatic when they are actually germline[32]. With high-depth sequencing, the estimated VAF will have a small error bar and the deviation from the expected 0.5 for germline variants will be easier to discriminate. Nonetheless, a large database of germline variants[33] including those from various ethnicities would be important to avoid

misclassification of germline as somatic variants. Improvements in these areas, along with very high-depth sequencing and carefully tuned analysis pipeline including annotation of LOD, will help deliver accurate mutational profiling data to help clinicians make optimal therapeutic decisions.

## Methods

**Study design.** Cancer patient samples were obtained at Samsung Medical Center from January 2014 to August 2016 (Supplementary Data 2) with informed consent from some patients and consent waived by the Institutional Review Board (IRB) for other patients. The IRB of Samsung Medical Center (SMC) approved this study. A pathologist examined each sample for diagnosis and tumor content. The inclusion criteria for specimens in this study are (i) the possibility that the patient could be enrolled in a clinical trial if an actionable mutation is discovered; and (ii) the patient's specimen was stored the pathology department with a sufficient amount of tumor fraction. Samples were typically processed without a paired normal tissue.

**DNA extraction and library preparation and sequencing.** Genomic DNA was extracted from fresh tissues using QIAamp DNA mini kits (Qiagen, Valencia, CA, USA) and from FFPE tissues using either a Promega Maxwell 16 CSC DNA FFPE kit or a QIAamp DNA FFPE Tissue kit. DNA concentration and purity were checked using a Nanodrop 8000 UV-Vis spectrometer (Thermo Scientific, Waltham, MA, USA) and Qubit 2.0 Fluorometer (Life Technologies, Grand Island, NY, USA). The degree of DNA degradation was measured using a 200 TapeStation Instrument (Agilent Technologies, Santa Clara, CA, USA) and real-time PCR (Agilent Technologies). Genomic DNA was sheared using a Covaris S220 (Covaris, Woburn, MA). Target capture was performed using the SureSelect XT Reagent Kit, HSQ (Agilent Technologies) and a paired-end sequencing library was constructed with a barcode. After checking for library quality, sequencing was performed on a HiSeq 2500 with 100-bp reads (Illumina, San Diego, CA, USA). For exome sequencing, SureSelect XT Human All Exon v5 was used for target capture; for whole-genome sequencing, the TruSeq Nano DNA Sample prep kit was used (Illumina).

**Panel design and sequencing.** Samples were profiled on CancerSCAN, a targeted-sequencing platform designed at Samsung Medical Center. This customized platform offered flexibility to include target genes curated from the literature of requested by the researchers and clinicians. These selected genes covered variants associated with the targeted cancer therapies (i) approved by the Korean MFDS and US FDA, (ii) in the clinical trials at the Precision Oncology Clinic at Samsung Medical Center or (iii) reported as having association with response of therapy in the public databases and the literature.

Using both existing and new algorithms (see below), we detected SNVs, small indels, CNVs, and gene fusions (see below). 1497 patients were profiled on CancerSCAN V1, which targeted 83 genes; the rest were profiled on Version 2, which targeted 381 genes. *TERT* promoter was also included. The genes contained in the two versions are listed in Supplementary Data 1.

**Purity estimation.** Computational inference of tumor purity from panel data is more difficult than from exome or whole-genome data since the limited set of genes is less likely to contain genomic alterations, which are informative for making inferences on tumor purity. First, we identify copy-neutral regions. The minor allele frequencies at known SNPs in the copy-neutral regions are near half and, when only those SNPs are considered, their read densities correspond to the most prominent peak in the distribution of read coverage at the SNPs, based on the hypothesis that pure polyploidy tumors with 4N in all chromosomes are extremely rare. The regions of copy number gain and loss are identified by their adjusted coverage relative to the copy number-neutral regions. Once the copy-neutral, gain, and loss regions are delineated, the following formula can be used to infer the proportion of each tumor clone:

$$\text{Alternative allele frequency} = \frac{P * Y + (1 - P)}{P * X + 2(1 - P)},$$

where $X$ and $Y$ are the numbers of all and alternative alleles at each group of clustered SNPs in the tumor, respectively, and P is a proportion of tumor clone ranging from 0 to 1. Tumor purity was inferred from the maximum value among the Ps estimated at multiple positions, with the hypothesis that the largest clone best represents tumor purity. Tumor purity below 30% was less reliable and was not annotated. More details will be available in a separate manuscript in preparation.

**Variant detection.** Alignment: The paired-end reads were aligned to the human reference genome (hg19) using BWA-MEM (v.0.7.5). SAMTOOLS (v0.1.18), GATK (v3.1-1), and Picard (v1.93) were used for file handling, local realignment, and removal of duplicate reads, respectively. We recalibrated base quality scores using GATK BaseRecalibrator based on known single-nucleotide polymorphisms (SNPs) and indels from dbSNP138.

SNV detection: To increase sensitivity, we used two published methods for SNV detection, MuTect[13] (v1.1.4) and LoFreq[14] (v0.6.1), with default parameters. The union of the variants identified by the two callers (with the high confidence (HC) set for MuTect) was used as the candidate set of variants. The number of false positives for the simulated cases in Fig. 3a is small, as shown in the estimated PPV curves (Supplementary Fig. 11a). Small indels were identified by Pindel[34] (v0.2.4) with its default setting. We applied several filtering steps to filter these putative germline variants: (i) variants with very high VAF (≥97%), except for the hotspot mutations; (ii) variants with population allele frequency >3% in the >400 normal samples in our database (this is important for removing ethnic-specific variants); and (iii) other frequently detected variants that are likely to be alignment artifacts or are in hard-to-sequence regions, as curated by manual review and compiled in our database. The variants were annotated by ANNOVAR[35].

We could improve the PPV of our SNV calling pipeline by an additional filtering step. MuTect already implements several steps for removing likely false positives, using features that are often characteristic of false positives such as proximity of an indel, multiple mutations in a small neighborhood, strand bias, and clustered read start/end positions. Briefly, our additional step implements a logistic regression model based on the factors mentioned above and trained it based on low VAF variants that arise in normal samples (in particular, a subset that has an abnormally high transition or transversion rate at each position, as they are likely to be enriched for false positives). We used 38 normal samples (28 normal tissues (Supplementary Data 2) and 10 normal HapMap cell lines (Supplementary Table 1)) profiled on the panel to train the classifier. We checked performance of the model by cross-validation and selected the parameter of the regression model by receiver operating characteristic (ROC) analysis of NA12878. When this model was applied to NA12878 (for which we have a well-validated true positive calls), we found that the number of false positives was substantially reduced (Supplementary Fig. 11 and Supplementary Data 7).

CNV detection: To identify somatic CNVs, we calculate the mean read depth at each exon, normalized by the coverage of the target regions in that sample. This normalized read depth is further standardized by dividing by the expected coverage for a normal individual (The expected coverage at each exon was taken to be the median of the read depth at that exon across a set of normal individuals). These steps account for the variability in capture efficiency and GC content at different exons. To infer the correct copy number, the amplitude of the copy numbers are then adjusted based on the estimated purity. If the adjusted amplitude of the copy change is greater than 1 or less than 1 (in log scale), the region is called as amplification or deletion, respectively.

Fusion detection: Most fusions involve intronic breakpoints. To identify fusion using a gene panel, we tiled across the "hotspot" introns that are known to contain most breakpoints for set of clinically relevant fusions. In version 2, introns of 22 genes were covered densely with capture probes. Since the average DNA fragment size was ~180 bp in our libraries (thus, with 100 bp reads, most fragments are fully sequenced), we expect each fusion to be reflected in multiple split reads. We require four split reads to make a fusion call, with at least two reads mapping to each side of the breakpoint. We also consider both primary and secondary alignments to increase sensitivity. Once the candidate fusions are identified, further filtering is performed using various features including mapping quality, insert size, CIGAR string, strand direction, alignment information, local cluster coverage, and concordance of the read alignment direction. The split reads allow mapping of the breakpoints with base pair resolution.

**Limit of detection estimation by manual dilution assay.** In conventional molecular tests, detection capability is typically specified in terms of LOD, following the guidelines established by The Clinical and Laboratory Standards Institute (CLSI), an international organization that develops clinical and laboratory practices[21]. American College of Medical Genetics and Genomics (ACMG) guideline for NGS also recommends that the lower LOD should be determined for variant detection based on dilution assays[22]. For a given sequencing depth, LOD, the lowest concentration that can be consistently detected with ≥95% sensitivity, is computed by (i) estimating sensitivity of detection at dilutions (in this case, variant allele fraction); and (ii) fitting a probit regression model to the data, which then allows estimation of the sequencing depth at which the sensitivity is 95%. (Probit regression is a linear regression model for a binary response variable and is similar to logistic regression except that the link function is the cumulative normal distribution function instead of logit).

We implemented a manual dilution assay, adopting the scheme used in Frampton et al.[36] Purified DNA of 10 HapMap cell lines (Supplementary Table 1) were obtained from Coriell Institute and were mixed in equal proportions. Each cell line and the mixed pool were sequenced using the CancerSCAN V2 panel. After SNPs were called for individual cell lines by both MuTect and LoFreq, more than ~1700 reliable exonic SNPs (those with a refSNP number) were identified, with a wide range of VAFs starting with ~5%, which corresponds to a unique heterozygous SNP in one cell line. Once the variants were called at these positions in the mixed pool, we could define true positives and false negatives as the variants detected and undetected, respectively, at various VAFs. To generate more data points at each VAF, we also down-sampled the pool at various fractions. These points were then used to construct the regression line and estimate the LOD at each VAF (Supplementary Fig. 12).

**Limit of detection estimation by in silico dilution assay**. In a manual dilution assay described above, the true concentration of each cell line in the mixed pool maybe different from the intended 10%. The deviation may arise due to inaccurate DNA quantitation and/or pipetting error, with greater deviation at low concentrations[37]. To avoid this problem, we also performed in silico dilution assay, using the well-characterized NA12878 cell line[38] sequenced on the CancerSCAN V2 panel.

We selected all heterozygous SNPs ($n = 222$) that have "high-confidence" genotype calls (version 2.19) and are in the target exonic region of the panel (Supplementary Fig. 13). To simulate various diluted VAFs at these heterozygous SNPs, we randomly selected reads containing variant alleles and replaced some of them with the reference bases, controlling the dilution level by the probability that the selected variant base will be converted, as illustrated in Supplementary Fig. 14. After a wide range of VAFs were obtained at the SNP positions, we followed the same procedure as in the manual assay to estimate LOD. This in silico approach resulted in similar sensitivity lines as in the manual assay, except that they were more stable for lower VAFs. Figure 3a is generated from this analysis.

**Performance comparison of SNV callers**. To set up an accurate SNV pipeline, we compared the performance of four callers (LoFreq, MuTect, Platypus[39], and VarDict[40]) and their combinations with default and modified parameters on 332 exonic SNPs (both heterozygous and homozygous SNPs in the target region) of NA12878 (Supplementary Data 7). The results involving LoFreq, MuTect, and their combinations on the in silico dilution data are shown in Supplementary Fig. 11.

**Digital PCR**. Digital PCR was performed on the QX200 ddPCR™ System (Bio-Rad) to validate low VAF (2.5–10.3%) variants in 59 samples (Supplementary Data 4). ddPCR reaction mixes were prepared with template gDNAs, ddPCR Supermix (Bio-Rad) and TaqMan primer-probe mixtures, and partitioned into oil droplets (~20,000) generated by the QX200 droplet generator. The droplets were then thermal-cycled using Veriti 96-Well Thermal Cycler (Life Technologies). Amplified droplets were imaged on the QX200 droplet reader (Bio-Rad), and analyzed by QuantaSoft™ software (Bio-Rad). The concentration of the nucleic acid sequence targeted by the FAM and VIC or FAM and HEX dye labeled probes was estimated using Poisson distribution.

**Classification of variants**. We classified all variants into three tiers (Supplementary Table 2). Tier 1 variants included the variants that were listed as therapeutic targets by the Korean MFDS or the US FDA as well as those that have been reported to be candidates for clinical trials (Supplementary Data 8). Both CancerSCAN V1 and V2 panels covered all positions of the tier 1 variants. Tier 2 variants included any mutation that was reported in COSMIC (version 64). For gene fusions, those involving a target gene and a known partner (in COSMIC) as Tier 1 and a novel partner as Tier 2. Tier 3 variants included all other mutations. "Actionable" variants in Fig. 5c are Tier 1 variants; "known" (but not actionable) alterations are Tier 2 variants. Most of our analysis deals only with the actionable (tier 1) variants that were in both versions. Some figures (Fig. 5c, Supplementary Fig. 4, and Supplementary Fig. 8) only include CancerSCAN V2 patients for a fair comparison of non-actionable proportions.

**Data availability**. The data that support the findings of this study are available on request from the corresponding author (W.-Y.P.). The raw data are not publicly available due retroactively collected samples not having explicit consent for sharing of raw data.

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

## Acknowledgements

This work was supported by the Korean Health Technology R&D Project, Ministry of Health & Welfare, Republic of Korea (HI13C2096 to W.-Y.P.). Korea Telecom (KT) Corporation supported server for computational analysis of public data.

## Author contributions

H.-T.S. and J.W.Y. performed analysis of all data, with guidance from Y.-L.C., P.J.P. and W.-Y.P. The manuscript was written by H.-T.S., J.W.Y. and P.J.P. H.-T.S., J.W.Y., N.K.D.K., J.-Y.N., C.L., D.R., S.C.K., K.P., E.L., D.S.S. and J.-G.J. created bioinformatics pipeline to support data analysis. S.-Y.K., H.J.J. and J.S.B. generated the genomic data. Y.-L.C., J.L., S.T.K., M.-J.A., S.-H.L., J.S.A., W.Y.L., B.Y.O., Y.H.P., J.E.L., K.H.L., H.C.K., K.-M.K., Y.-H.I. and K.P. provided clinical data. All authors reviewed the manuscript.

## Additional information

**Competing interests:** The authors declare no competing financial interests.

