## [Peer Review File · Nature Communications]

Reviewers' comments:

Reviewer #1 (Remarks to the Author):

The manuscript titled "Prevalence and detection of low-allele-fraction variants in clinical cancer samples" by Dr. Shin et al reported tumor profiling data generated by gene-panel sequencing of >5,000 clinical samples in one hospital. By tabulating the variant allele frequency (VAF) distribution based on hotspot single nucleotide variants (SNVs), the authors emphasized the need to have deep sequencing by gene panel to ensure sufficient power for detecting low-frequency variants. To demonstrate the importance of detecting low VAF mutations, the authors presented a comparison of the estimated tumor purity in their clinical samples with those used for research investigation by The Cancer Genome Atlas Project (TCGA). To further illustrate the clinical importance of low VAF variants, the authors presented two cases for whom partial remission was achieved upon receiving targeted therapy based on low VAF variants although response for one patient was not durable.

At introduction, the manuscript highlighted a broad spectrum of issues associated with clinical sequencing. However, the presentation focused on low VAF hotspot mutations, which does not add much new insight into the general knowledge that low tumor purity and tumor heterogeneity would require deep sequencing to ensure sensitivity. Specific issues are outlined as follows.

Major issues:

1. The analysis lacked the sophistication to delineate the sample heterogeneity and assay heterogeneity of this large clinical cohort. Table S2 and various sections of the paper shown inclusion of refractory cases which may have treatment induced mutations (approximately 50% of the samples) and an example presented in Fig. S4 indicated that a subset of tumors were paired as both prior and post- treatment specimen were analyzed. The overall statistics presented in Fig. 1 did not stratify based on tumor acquired at diagnosis or post-treatment. On the other hand, the treatment induced mutations were discussed throughout the manuscript anecdotally not systematically. Further, there are two gene panels, one with 83 genes and the other with 381 genes, which could potentially complicate data harmonization. This issue was not addressed at all in the manuscript. Sample stratification based on prior or post treatment would have improved the quality and relevance of many analysis, e.g. tumor purity comparison with TCGA.
2. Novelty of the study. The low VAF distribution of mutation hotspots was described previously by Frampton et al 2013, therefore the current work appeared to be an extension of the 2013 study. The concept of "limit-of-detection" was also previously introduced by Frampton et al in Fig. 2b of their paper.
3. VAF distribution was shown using SNV markers. On the other hand, indel analysis was mentioned a various places but without a summary statistics. Detailed analysis on indel would be of higher interest as custom capture by gene-panel would have reduced efficiency in capturing the indel variant. This is particularly pertinent for the EGFR because exon 19 indels are known therapeutic targets for kinase inhibitors. Fig. 1a did not include any of the exon 19 indels in the analysis; however in the 8-sample analysis of EGFR C797S mutation in one of the supplementary figures, the authors did present exon 19 deletion. It is obvious that the authors recognized the importance of exon 19 indels therefore it was quite puzzling why indels are not presented in Fig. 1a.
4. The authors noted about the limitation of the gene-panel sequencing in detecting copy number and gene fusions. Based on distribution in Fig. S9, there should be several cohorts with extensive whole-genome sequencing data published by the consortiums like TCGA. How does the SNV/indel, CNV and fusions derived from gene panels of their study compare with those from the TCGA cohort of the same cancer type? Was the difference caused by technical issues or reflect the sample differences?
5. The outcome for the two cases who received targeted therapy based on low VAF mutations is interesting. It will be good if the authors can collect more data to assess whether these were expected or exceptional response. Further, the fact that the response in the second case was not durable would also require discussion regarding the effectiveness of targeted therapy in low VAF.

Minor issues:

1. The germline controls used for this study. The authors used a mixture of panel sequencing of cell lines, whole genome and whole exome sequencing to serve as a control for removal of germline variants in tumor. Why did not use a reference germline cohort that was analyzed entirely by the gene panel? It would also be good for the authors to review the proportion of germline mutations that were misclassified as somatic as described by Jones et al (2015) which reported major issues with tumor-only sequencing. Selecting a subset of cases for germline sequencing will be a good way to answer that question.
2. Figure. S4b. What is the leftmost variant? Is it relevant? This one also appeared to be enriched after treatment. Is this a biologically relevant variant?
3. line 408-410, the authors mentioned about training their filtering method by using "low VAF variants that arise in normal samples (in particular, a subset that has an abnormally high transition or transversion rate at each position, as they are likely to be enriched for false positives)." Are these "normal samples" from the 10 cell lines that were subjected for panel sequencing? Are the size of these samples sufficient? The authors mentioned "The parameters were obtained by cross-validation.". What does that mean?

Reviewer #2 (Remarks to the Author):

Shin et al. describe a study characterizing frequent low variant allele fractions (VAFs) of clinically actionable mutations in a panel of over 5,000 tumor samples, leading to recommendations for best practices in clinical use of tumor sequencing. This is an interesting and potentially valuable paper, despite some speculative points with regard to impact. The paper makes a compelling case for the importance of the question of how to use sequencing reliably for clinical decision making in cancer treatment. It produces some potentially valuable concrete recommendations, particularly with regard to standards for read depths and the importance of limit of detection (LOD) assays in bringing sequencing to clinical practice. The work might have been stronger with more definitive evidence that better characterizing low-VAF variants, via the recommended practices, will lead to better clinical outcomes. The argument that they should is, however, logical and plausible, if unproven. The methodology behind the paper is, with minor exceptions, well thought out and nicely validated. I have only a few comments on the details of the methodology, since the authors have done an excellent job of anticipating and preparing for potential criticisms.

My most substantial critique is simply that the case is not compelling made for the clinical impact of discovering the extra low-VAF actionable variants one would find by following this paper's recommendations. The paper presents solid evidence that these low-VAF actionable variants exist and are fairly common. But only a couple of anecdotes are given for the effects of acting on such variants, and with mixed success. While it is certainly plausible that finding and acting on more low-VAF variants, as proposed here, will lead to lives lengthened or saved, a serious clinical or at least animal study would be needed to prove that. It is reasonable to argue that that would be beyond the scope of the present work, and the paper is still interesting and valuable without it. But that lack does make this a much less significant paper than it might be.

A more minor issue is that some aspects of purity estimation could use better support. Using of 50% minor allele frequency as the standard for detecting copy number neutral regions is not entirely convincing, since one can end up with 50% minor allele frequency (MAF) in an amplified region. Sequential duplication of both alleles, genome duplication, or just bad luck with relative frequencies of different clones could all plausibly yield approximately 50% MAF in a non-diploid genomic region. Perhaps these alternatives are rare enough to allow one to discount them, but it would be useful to see some evidence or at least mathematical justification to argue that. The formula for tumor purity (between lines 381 and 382) also raises some question, in that it seems to rely on the assumption that one is dealing with a single clone and are not obviously valid in the presence of intratumor heterogeneity. The decision to take the maximum estimate over several at multiple positions may be a way of correcting for such effects, but if so, that should be argued.

While the main points of the paper do not require that purity estimates are accurate, putting them into practice would seem to require better evidence that they are accurate or at least conservative. Deferral of details to another manuscript in preparation makes this section difficult to critique fairly.

Finally, it would be useful to see some brief statement of the criteria used to derive the 381 cancer related genes in the analysis panel (Table S1). The gene list itself has no obvious problems, but it would be useful to see some argument that it is a reasonably representative and unbiased list of genes one might wish to profile in similar clinical applications of tumor sequencing.

Response to Reviewers for Shin et al.

We thank the reviewers for their detailed comments. They have made a number of excellent points, which have helped us greatly in improving the manuscript.

Both reviewers pointed out a major shortcoming of our previous version, which is that we did not have treatment information for the patients. As those who have tried to curate clinical data from a hospital data system can appreciate, gathering clinical information with respect to treatment is difficult for many reasons, even with electronic medical records. For instance, access to clinical record was restricted to physicians in our system, which meant that non-physician researchers could not access data and had to rely on physicians for curation; when the samples were received for sequencing, the collection date was often missing, thus requiring an individual follow-up in each case for that information with dozens of physicians.

Nonetheless, to respond to the reviewers' comment, we were able to review the medical charts for ~1600 patients to determine whether the sampling was done pre- or post-treatment. This allowed us to perform several new analyses as described below. This has improved the manuscript significantly and so we thank the reviewers for pushing us to do this.

We have also modified the manuscript substantially to address all of the reviewers' comments, including more discussion of indels; discussion of two panel versions; more details on copy number and purity estimation; clarification of our contribution compared to Frampton et al; additional data on germline filtering criteria; and comparison of variant allele frequencies with TCGA data.

Reviewer #1 (Remarks to the Author):

The manuscript titled "Prevalence and detection of low-allele-fraction variants in clinical cancer samples" by Dr. Shin et al reported tumor profiling data generated by gene-panel sequencing of >5,000 clinical samples in one hospital. By tabulating the variant allele frequency (VAF) distribution based on hotspot single nucleotide variants (SNVs), the authors emphasized the need to have deep sequencing by gene panel to ensure sufficient power for detecting low-frequency variants. To demonstrate the importance of detecting low VAF mutations, the authors presented a comparison of the estimated tumor purity in their clinical samples with those used for research investigation by The Cancer Genome Atlas Project (TCGA). To further illustrate the clinical importance of low VAF variants, the authors presented two cases for whom partial remission was achieved upon receiving targeted therapy based on low VAF variants although response for one patient was not durable.

At introduction, the manuscript highlighted a broad spectrum of issues associated with clinical sequencing. However, the presentation focused on low VAF hotspot mutations, which does not add much new insight into the general knowledge that low tumor purity and tumor heterogeneity would require deep sequencing to ensure sensitivity. Specific issues are outlined as follows.

We agree that the general point that deep sequencing is required to ensure sensitivity is familiar to anyone doing sequencing. The main contribution is the extent to which patients with clinically meaningful low VAF variants are present in the clinic. A very practical question in implementing

clinical sequencing in the hospital was how many people would benefit if sequencing is done at a higher depth. We are not aware of another paper that has examined a large number of clinical cases; we also show, as the reviewer noted, that opining on the appropriate sequencing depth based on TCGA data would be misleading.

On a related note, we focused on the hotspot mutations because the focus of the study was not a general description of overall heterogeneity but how heterogeneity affects treatment decisions. As such, we were interested mainly in the variants that are (or may soon be) actionable. We have modified the text to better describe our focus.

Major issues:

1. The analysis lacked the sophistication to delineate the sample heterogeneity and assay heterogeneity of this large clinical cohort. Table S2 and various sections of the paper shown inclusion of refractory cases which may have treatment induced mutations (approximately 50% of the samples) and an example presented in Fig. S4 indicated that a subset of tumors were paired as both prior and post-treatment specimen were analyzed. The overall statistics presented in Fig. 1 did not stratify based on tumor acquired at diagnosis or post-treatment. On the other hand, the treatment induced mutations were discussed throughout the manuscript anecdotally not systematically. Further, there are two gene panels, one with 83 genes and the other with 381 genes, which could potentially complicate data harmonization. This issue was not addressed at all in the manuscript. Sample stratification based on prior or post treatment would have improved the quality and relevance of many analysis, e.g. tumor purity comparison with TCGA.

The reviewer brings up two important issues here.

1) Lack of stratification between pre- and post-treatment cases.

As mentioned above, we have used the additional clinical annotation in several additional analyses. Specifically, we could classify 1557 samples as pre or post-treatments (pre:1203, post: 354).

In general, the VAF distributions of actionable mutations do not show substantial difference between the two groups (See Fig), although they tends to be a little lower post-treatment. Tumor purity also did not show significant difference.

In some specific cases, we do observe a statistically significant difference. In post EGFR- tyrosine kinase inhibitor (TKI) therapy samples, for instance, we expect to find tumor resistance-inducing

Fig. 8. Comparison of VAF distribution of actionable SNVs from samples classified pre and post treatment (chemotherapy). (a) VAF distributions of each class are shown after kernel density smoothing and there are no significant difference of distribution between the classes ($p > 0.05$, wilcoxon rank sum test). The mutations that were observed more than once are summarized here. (b) VAF distributions of individual actionable variants.

clones containing secondary mutations after treatment. We could identify 141 lung cancer cases sampled prior (n=31) or post (n=110) EGFR-TKI therapy (afatinib, erlotinib or gefitinib). In these samples, we found that the VAF distributions at the hotspots were significantly different ($p=0.018$) mostly due to *EGFR* T79M, although tumor purity was not (see figure below).

Fig. 5. Comparison of VAF of *EGFR* variants and tumor purity of lung cancer samples (n=141) based on pre and post EGFR-TKI therapy (afatinib, erlotinib, or gefitinib). (a) Comparison of VAF. (b) Comparison of tumor purity. The p values were calculated using the wilcoxon rank sum test. ns, not significant.

To determine whether the mutation count was changed after chemotherapy, we compared variant counts of each cancer type (only V2 sample included) with stratification (see figure below). The result has a limitation because it is compared only in the panel target area, but we could see that there was no significant difference overall.

Fig. 5. Comparison of variant count of each cancer type classified pre and post treatment (chemotherapy). The p values were calculated using the wilcoxon rank sum test. ns, not significant.

2) **Differences between two panel versions.** The reviewer brought up the issue of compatibility between the two versions, as we have increased the number of genes from 80 to 380 for the second version. We agree that this is an area that generally requires a careful analysis—when you put together a database like ExAC, there are numerous issues associated with combining multiple capture technologies, different target regions, and sequencing depths, etc. Fortunately, this is a minor issue for our manuscript because our analysis deals only with the clinically important hotspot mutations that were in both versions (Tier 1; see the “Classification of variants” section in Methods). Some improvement in purity estimation does come with a larger target region, but it is not significant enough to alter any results. We acknowledge that we should have described this issue more in the manuscript, and have modified it accordingly.

2. Novelty of the study. The low VAF distribution of mutation hotspots was described previously by Frampton et al 2013, therefore the current work appeared to be an extension of the 2013 study. The concept of “limit-of-detection” was also previously introduced by Frampton et al in Fig. 2b of their paper.

We agree with the comment that the detection sensitivity portion of our paper is an extension of Frampton et al. However, we think that our paper has many new important contributions even on that issue.

- Frampton et al actually does not describe the distribution of VAF in patient samples. Their “expected allele frequency” (their Figure 2a) is from their simulation designed simply to represent a broad range of allele frequencies. Ours is the first to describe the distribution in a large patient population.
- Frampton et al consider 5% as their lowest limit, and do not mention hotspots that may have very low VAFs, e.g., *EGFR* T790M and T797S (often found in less than 5%). Our Figure 1 illustrates the importance of such cases in a clinical setting.
- The concept of “limit-of-detection” is actually very old, going back at least to the 1980s, when LOD was defined in analytical chemistry as the lowest quantity or concentration of component that can be reliably detected with a given method. This seems like a straightforward concept, almost too simple to even highlight for bioinformaticians. But those in clinical diagnostics are more familiar with this (though they do not explicitly note their definition of sensitivity; we used 95% here), as evidenced by its usage in the American College of Medical Genetics and Genomics NGS Guidelines. Therefore, we want to mention this term and highlight its utility as a measure of an NGS assay in the context of other non-NGS clinical assays. Also, although the dilution-based coverage vs sensitivity plot across a wide range of coverage in Frampton et al is useful, many others have drawn similar sensitivity plots, though based on simpler simulations.

3. VAF distribution was shown using SNV markers. On the other hand, indel analysis was mentioned a various places but without a summary statistics. Detailed analysis on indel would be of higher interest as custom capture by gene-panel would have reduced efficiency in capturing the indel variant. This is particularly pertinent for the EGFR because exon 19 indels are known therapeutic targets for kinase inhibitors. Fig. 1a did not include any of the exon 19 indels in the analysis; however in the 8-sample analysis of EGFR C797S mutation in one of the supplementary figures, the authors did present exon 19 deletion. It is obvious that the authors recognized the importance of exon 19 indels therefore it was quite puzzling why indels are not presented in Fig. 1a.

This is an excellent point. We focused on SNVs because, as the reviewer knows well, accurate identification of indels is harder than it is for SNVs. Even without the differences in capture efficiency, trickier alignment and other issues make it more challenging to analyze indels.

But, in response to the reviewer's comment, we have analyzed indels further and now include a new figure for Tier 1 indel (see figure below). The fractions of indels <5%, <10%, and <20% are ~18% (50/274), ~30% (83/274), and ~50% (138/274).

Fig. S. VAF distributions for actionable indels in our cohort. (a) Histogram of VAFs in actionable indels. The Tier 1 indels that were observed more than once are summarized here. **(b)** VAF distributions of individual actionable indels.

4. The authors noted about the limitation of the gene-panel sequencing in detecting copy number and gene fusions. Based on distribution in Fig. S9, there should be several cohorts with extensive whole-genome sequencing data published by the consortiums like TCGA. How does the SNV/indel, CNV and fusions derived from gene panels of their study compare with those from the TCGA cohort of the same cancer type? Was the difference caused by technical issues or reflect the sample differences?

This is an interesting but a difficult one to address in general because there are many variables to consider. We compiled a table (below) that compares the number of actionable mutations and their types in five tumor types with large number of cases in this study vs TCGA. The numbers are mostly comparable, e.g., 57% vs 55% for the proportion of breast cancer cases with actionable SNVs or CNVs. For pancreatic cancer and gastric cancers, the numbers were more different, but this is likely due to the less frequent *KRAS* mutations in the Korean population. But it is not clear exactly what we can conclude from the table since the populations that two studies are different, with TCGA patients being mostly Caucasians, whereas those in this study almost exclusively Korean).

As to the impact of the differences in platform (panel vs exome/genome), we would like to refer to a paper that we have published recently (Yang et al, Analyzing somatic genome rearrangements in human cancers by using whole-exome sequencing, *AJHG*, 2016), where we tried to detect fusion genes across ~5000 TCGA exomes. To study the difference in sensitivity between the exome-based and WGS-based calls, we examined 120 cases that were profiled on both. As shown in panel C below (figure taken from the above paper), the total number of fusions detected from WGS is obviously much greater (yellow circle). A good number of fusions with exonic breakpoints (31) or non-exonic breakpoints (31) are captured by both exomes and WGS. But interestingly, 114 fusions with exonic breakpoints are found by WGS-only, while 40 are captured by exomes only. We think the 114 are missed on exomes due to the target design/capture inefficiency, whereas the 40 are missed on WGS due to lower coverage. Panel B shows a case where the breakpoint is detected on both platforms; Panel D shows a case of low VAF fusion discovered only by exomes. Another variable in this analysis is the depth of sequencing: TCGA genomes had 30-60X coverage (depending on when they were sequenced) and exomes were ~100X on average. Yet another variable is the detection algorithm, but we think that we have done a fairly good job of detecting them using our custom algorithm, which we was optimized after extensive PCR validations (this is the algorithm we used to call the fusions in ~dozen TCGA marker papers). Therefore, for our panel-based data, we suspect that our detection sensitivity for the clinically important fusions that we targeted is high, as the target breakpoint regions were densely tiled (sensitivity and specificity were also validated by standard PCR-based clinical assays). For fusions that were not targeted or targeted but had breakpoints in less common introns, the sensitivity of our panel is obviously lower.

For CNVs, it is well known that exome-based (and panel-based) estimates are generally less reliable and are heavily dependent on the algorithm. We showed this recently (Nam et al, Evaluation of somatic copy number estimation tools for whole-exome sequencing data, *Briefings in Bioinformatics*, 2016), where we compared exome-based calls against the consensus calls from WGS and SNP arrays. However, for our panel data, we are interested in high copy number changes in specific regions and our calls have been compared with PCR-based calls. Where we suspect a moderate gain in copy number and tumor purity cannot be easily determined, the call is harder to make. Because of there are many parameters involved, it is difficult to determine how much of there differences are technical in nature. There is more discussion of CNVs in our response to Reviewer 2.

Table S (below). Comparison between samples with panel sequencing and TCGA data. We selected five major tumor types for which the sample sizes were large in our data. For TCGA fusions (those for which probes are present on the panel), we obtained the counts from RNA-seq-based database.

Tumor type		This study	TCGA cohort
Breast cancer	Proportion of sample with actionable target (≥ 1)	57% (N=575)	55% (N=963)
	Number of all actionable mutation	448	603
	Proportion of actionable SNV	38%	45%
	Proportion of actionable CNV	62%	55%
	Proportion of actionable fusion	0%	0%
Colorectal cancer	Proportion of sample with actionable target (≥ 1)	59% (N=509)	66% (N=220)
	Number of all actionable mutation	375	153
	Proportion of actionable SNV	91%	89%
	Proportion of actionable CNV	9%	11%
	Proportion of actionable fusion	0%	0%
Lung cancer	Proportion of sample with actionable target (≥ 1)	51% (N=910)	68% (N=230)
	Number of all actionable mutation	741	171

	Proportion of actionable SNV	71%	63%
	Proportion of actionable CNV	28%	34%
	Proportion of actionable fusion	1%	4%
Pancreatic cancer	Proportion of sample with actionable target (≥ 1)	72% (N=71)	91% (N=149)
	Number of all actionable mutation	56	154
	Proportion of actionable SNV	95%	88%
	Proportion of actionable CNV	5%	12%
	Proportion of actionable fusion	0%	0%
Gastric cancer	Proportion of sample with actionable target (≥ 1)	26% (N=949)	44% (N=393)
	Number of all actionable mutation	297	184
	Proportion of actionable SNV	51%	47%
	Proportion of actionable CNV	49%	53%
	Proportion of actionable fusion	0%	0%

5. The outcome for the two cases who received targeted therapy based on low VAF mutations is interesting. It will be good if the authors can collect more data to assess whether these were expected or exceptional response. Further, the fact that the response in the second case was not durable would also require discussion regarding the effectiveness of targeted therapy in low VAF.

We have collected more survival data to address the reviewer's question, focusing on *EGFR* T790M and L858R/Exon19del to ask whether the low VAF cases ($\leq 5\%$) have distinct Kaplan-Meier curves compared to the high VAF cases ($> 5\%$). These results show that there is no significant survival difference between the patients with low vs high VAF variants who receive targeted therapy (Fig S-1). To show that this result does not depend on a specific threshold for defining "low" vs "high" AF, we also tried all possible cuts into two groups and computed the p-value in each case (Fig S-2). We find that no comparison of survival data are significant, strongly suggesting that detection of low VAF is important for clinical decision-making.

Fig. S-1. Survival comparison of lung cancer patients with EGFR-TKI therapy according to VAF of EGFR mutations. (a) Progression free survival comparison of lung cancer patients (n=46) receiving 3rd generation EGFR-TKI therapy with EGFR T790M and without EGFR CNV. (b) Survival analysis of stage III and IV lung cancer patients (n=66) receiving 1st or 2nd generation EGFR-TKI therapy with EGFR activation variants and without EGFR T790M and EGFR CNV.

Fig. S-2. p values of log-rank test calculated using each VAF of hotspot variants as a cutoff. (a) p values of samples of Fig S-1a. (b) p values of samples of Fig S-1b.

Minor issues:

1. The germline controls used for this study. The authors used a mixture of panel sequencing of cell lines, whole genome and whole exome sequencing to serve as a control for removal of germline variants in tumor. Why did not use a reference germline cohort that was analyzed entirely by the gene panel? It would also be good for the authors to review the proportion of germline mutations that were misclassified as somatic as described by Jones et al (2015) which reported major issues with tumor-only sequencing. Selecting a subset of cases for germline sequencing will be a good way to answer that question.

This is an important issue, one that we have examined carefully. We are certainly aware of the Jones et al paper, but we think that paper has an inflated rate of germline variants that get misclassified as somatic—in our opinion, they, perhaps in an attempt to highlight the problem of tumor-only sequencing, do not do as thorough a job in filtering for potential germline variants as some others have done, in our opinion. Another paper (Garofalo et al, *Genome Medicine*, 2016) found that their FP rate is 14% and that 93% of these FP variants were later interpreted as “uncertain variants” by pathologists. As we had mentioned in our manuscript already, another paper (Hiltemann et al, *Genome Research*, 2015) showed that, with ~400 unrelated genomes for filtering, somatic SNVs could be distinguished from germline variants as efficiently as having matched normals.

Indeed, we had performed the experiment that the reviewer suggested (we hadn’t included this in the paper before, but we have added this to the revised version). Briefly, we had sequenced 74 breast cancer samples with matched normals (tumor: panel at ~900X, normal: WES at ~100X) (see table below). To test how efficient our germline filtering strategy is for tumor-only sequencing, we put aside the matched normals and applied these steps in our standard pipeline:

- Variants with very high VAFs ($\geq 97\%$) are filtered out except for hotspot mutations.
- Variants with population allele frequency $> 3\%$ in the > 400 normal samples in our database (this is important for removing ethnic-specific variants)
- Other frequently detected variants that are likely to be alignment artifacts or are in hard-to-sequence regions, as curated by manual review and compiled in our database.

After these filtering steps, only 5.5% (112/2040) of the germline variants (identified from WES; see the table below) remained, and ~10% (112/1082) of the variants we identified would have been germline. There are also two likely somatic variants that are mistakenly filtered out (the variant is removed from the tissue but was not present in blood). We will add the more details of filtering step in the method section.

Filtering step	No. of variant from panel seq	No. of blood germline from WES	No. of real somatic variant (Panel - WES)
All variants	3012	2040	972
filtering $\geq 97\%$ VAF	2205	1233	972
filtering using custom database	1082	112	970

2. Figure. S4b. What is the leftmost variant? Is it relevant? This one also appeared to be enriched after treatment. Is this a biologically relevant variant?

The leftmost variant in S4b is a synonymous SNP (rs1050171). It appeared to have been enriched after treatment only because the screenshots could only show a top portion of the full read view (we do not enough space to show $> 1000X$ at the position). But, as you can see below, this germline variant exists before and after treatment.

pre AZD9291 treatment:

post AZD9291 treatment:

3. line 408-410, the authors mentioned about training their filtering method by using “low VAF variants that arise in normal samples (in particular, a subset that has an abnormally high transition or transversion rate at each position, as they are likely to be enriched for false positives).” Are these “normal samples” from the 10 cell lines that were subjected for panel sequencing? Are the size of these samples sufficient? The authors mentioned “The parameters were obtained by cross-validation.”. What does that mean?

We use 38 normal samples to train the classifier (10 cell lines + 28 normal samples profiled on the panel). There were enough positions for training (positive: 15459, negative: 74921). We will add the information to Methods description. By “parameters”, we meant the prediction threshold based on our regression model, as chosen by our ROC analysis of NA12878. We have clarified this sentence in the Methods.

Reviewer #2 (Remarks to the Author):

Shin et al. describe a study characterizing frequent low variant allele fractions (VAFs) of clinically actionable mutations in a panel of over 5,000 tumor samples, leading to recommendations for best practices in clinical use of tumor sequencing. This is an interesting and potentially valuable paper, despite some speculative points with regard to impact. The paper makes a compelling case for the importance of the question of how to use sequencing reliably for clinical decision making in cancer treatment. It produces some potentially valuable concrete recommendations, particularly with regard to standards for read depths and the importance of limit of detection (LOD) assays in bringing sequencing to clinical practice. The work might have been stronger with more definitive evidence that better characterizing low-VAF variants, via the recommended practices, will lead to better clinical outcomes. The argument that they should is, however, logical and plausible, if unproven. The methodology behind the paper is, with minor exceptions, well thought out and nicely validated. I have only a few comments on the details of the methodology, since the authors have done an excellent job of anticipating and preparing for potential criticisms.

My most substantial critique is simply that the case is not compelling made for the clinical impact of discovering the extra low-VAF actionable variants one would find by following this paper's recommendations. The paper presents solid evidence that these low-VAF actionable variants exist and are fairly common. But only a couple of anecdotes are given for the effects of acting on such variants, and with mixed success. While it is certainly plausible that finding and acting on more low-VAF variants, as proposed here, will lead to lives lengthened or saved, a serious clinical or at least animal study would be needed to prove that. It is reasonable to argue that that would be beyond the scope of the present work, and the paper is still interesting and valuable without it. But that lack does make this a much less significant paper than it might be.

We thank the reviewer for his positive comments.

We absolutely agree with the reviewer's point that it would have been better to present a more outcome-driven analysis for the therapeutic benefits in low-VAF cases. As you may have seen in our response to reviewer 1's comment #5, we now use the treatment/survival data that we have curated to show that the low-VAF cases do not appear to behave differently from the rest in terms of survival. Admittedly, we wish we could have been more comprehensive with respect to the mutations considered for this type of analysis. But, as the reviewer is well aware, getting a sufficient number of cases with full treatment and survival data is difficult in these studies, partly because many patients are still alive and partly because the variation in treatment regimen and other factors make it hard to remove potential confounders.

We do think that our study presents a significant step forward, no doubt to be complemented by many future studies on the impact of low-VAF mutations in key genes.

A more minor issue is that some aspects of purity estimation could use better support. Using of 50% minor allele frequency as the standard for detecting copy number neutral regions is not entirely convincing, since one can end up with 50% minor allele frequency (MAF) in an amplified region. Sequential duplication of both alleles, genome duplication, or just bad luck with relative frequencies of different clones could all plausibly yield approximately 50% MAF in a non-diploid genomic region. Perhaps these alternatives are rare enough to allow one to discount them, but it would be useful to see some evidence or at least mathematical justification to argue that.

We agree with your points—it would be great if there is a better way to get around this problem. When there is whole-genome duplication in even number (mostly 4N; 6N or greater seems to be extremely rare in our experience), all the MAF would be around 50% and it would be difficult to calculate purity/ploidy accurately, unless there is LOH. In fact, this is a problem even for WES and WGS data, because different combinations of purity and ploidy sometimes appear to be compatible with the data. In fact, we have found that available algorithms often give discrepant results even for WGS (and we have a manuscript that addresses this). Therefore, in our purity estimation, we only included samples with deletion, amplification in odd numbers, or LOH. For a large minority of cases, we could not reliably infer purity, and those samples were left out from relevant analysis. We also note that we have reviewed the data manually in a substantial number of less clear cases, and calls were made based on the experience of the pathologists (two of the lead authors are pathologists).

We have validated the robustness of our approach with *in silico* cell line mixing experiments. Briefly, we produced panel sequencing data from three cancer cell lines (HCC2218, NCI-H2009, NCI-H2122) as well as from a pair of normal cell lines. Then we produced tumor-normal mixtures, with the tumor fraction ranging from 5% to 95% with a 5% increment. Then, we compared the inferred purity and reference purity. We find that our method gives good concordance (Panel A). Additionally, we compared inferred purity values with histological purity values generated by cell counting from two pathologists (Panel B). Although there is a wide variation in inferred purity for each partition of the histological purity, this may be in large part due to the inaccuracies in the histological purity estimation (e.g., the tumor portion viewed by the pathologist is not the same as the portion sequenced). Importantly, we note that our panel-based estimates are not worse than those by ABSOLUTE, a popular purity estimation method for exome data, as shown in Panel C (taken from Figure 2 in Carter et al, Absolute quantification of somatic DNA alterations in human cancer, *Nature Biotech*, 2012). Finally, in panel D, we show how the copy number estimates for some example genes stay nearly constant even as the purity changes in our mixtures (dotted lines shows what the estimates are when purity is not considered; the solid lines show the purity-corrected estimates).

The formula for tumor purity (between lines 381 and 382) also raises some question, in that it seems to rely on the assumption that one is dealing with a single clone and are not obviously valid in the presence of intratumor heterogeneity. The decision to take the maximum estimate over several at multiple positions may be a way of correcting for such effects, but if so, that should be argued.

While the main points of the paper do not require that purity estimates are accurate, putting them into practice would seem to require better evidence that they are accurate or at least conservative. Deferral of details to another manuscript in preparation makes this section difficult to critique fairly.

Again, we agree with the reviewer's incisive comment that tumor heterogeneity could potentially affect our purity estimates. Given that estimating the presence of multiple subclones and their fractions is difficult even for WGS data, a generalizable solution for panel data does not seem feasible, as the reviewer clearly understands. Therefore, when several regions in a tumor sample have CN alteration or LOH, we infer subclonal proportion in each region (See Methods). Then, with the assumption that the largest subclonal proportion represents the earliest CN alteration or LOH during tumor evolution, we take the largest value as the best approximation of the tumor purity. We do think that our taking the most major clone to calculate tumor purity is a conservative approach, although it could be lead to underestimation of purity in some samples. We hope that the data presented in the figure above is satisfactory to the reviewer.

Finally, it would be useful to see some brief statement of the criteria used to derive the 381 cancer related genes in the analysis panel (Table S1). The gene list itself has no obvious problems, but it would be useful to see some argument that it is a reasonably representative and unbiased list of genes one might wish to profile in similar clinical applications of tumor sequencing.

There was already a statement about the gene selection in the "Panel design and sequencing" section in method, but we have added more information to the section.

REVIEWERS' COMMENTS:

Reviewer #2 (Remarks to the Author):

As with the first draft, I believe this is an interesting and valuable paper that addresses some important questions on the path from current understanding of tumor heterogeneity and progression to translational impact. I will refrain from restating in detail the positive points of the work discussed in my critique of the previous submission. The revision and response to reviewers do a very good job of addressing the concerns raised to that prior submission. The authors offer a reasonable answer to the challenges of purity estimation, and the decision to omit samples for which it could not be determined reliably is appropriate. The additional data validating the present purity approach is persuasive. Likewise, there is a reasonable justification offered for the strategy of estimating purity when multiple genomic regions yield differing estimates. Other points of confusion are reasonably clarified to my satisfaction. While the case for clinical relevance is still not proven, the demonstration of similar survival for low-*VAF* vs. high-*VAF* variants does strengthen it. It is fair to defer any direct demonstration of clinical impact to future work. I will leave it to Reviewer 1 to decide if the responses to his or her critiques were persuasive, but they raise no new concerns for me. I have no further criticism to offer at this point.